# Cryptococcal Meningoencephalitis in Phenotypically Normal Patients

**DOI:** 10.3390/pathogens12111303

**Published:** 2023-10-31

**Authors:** Pia M. Cumagun, Mary Katherine Moore, Todd P. McCarty, Gerald McGwin, Peter G. Pappas

**Affiliations:** 1Department of Medicine, Division of Infectious Diseases, The University of Alabama at Birmingham, Birmingham, AL 35294, USA; piamariea.cumagun@uabmc.edu (P.M.C.);; 2School of Medicine, University of South Alabama, Mobile, AL 35233, USA; 3Department of Epidemiology, School of Public Health, University of Alabama, Birmingham, AL 35294, USA

**Keywords:** cryptococcal meningoencephalitis, immunocompetent hosts, PIIRS (post-infectious inflammatory response syndrome)

## Abstract

Cryptococcosis is an invasive fungal infection found worldwide that causes significant morbidity and mortality among a broad range of hosts. There are approximately 223,000 new cases of cryptococcosis annually throughout the world, and at least 180,000 deaths are attributed to this infection each year. Most of these are due to complications of cryptococcal meningoencephalitis among HIV-infected patients in resource-limited environments. The majority of individuals diagnosed with cryptococcosis have underlying conditions associated with immune dysfunction such as HIV, solid organ transplant, hematologic malignancy, organ failure syndromes, and/or the use of immunosuppressive agents such as glucocorticosteroids and biologic agents. In most clinical series, there is a small proportion of patients with cryptococcosis who are phenotypically normal; that is, they have no clinically obvious predisposition to disease. Cryptococcal meningoencephalitis (CME) presentation and management differ substantially between these normal individuals and their immunocompromised counterparts. In this review, we will focus on CME in the phenotypically normal host and underscore differences in the clinical presentation, management, outcome, and potential risk factors for these patients compared to immunocompromised persons who develop this potential devastating invasive fungal infection.

## 1. Introduction

Cryptococcosis is an important opportunistic fungal infection causing considerable morbidity and mortality among immunocompromised patients, including those positive for human immunodeficiency virus (HIV), organ transplant recipients, those who receive chronic glucocorticosteroids, recipients of biologic disease-modifying agents, hematologic malignancies, sarcoidosis, and other disorders characterized by dysfunction of cell-mediated and/or humoral immunity [1,2]. Not uncommonly, cryptococcosis affects individuals with no obvious immunocompromising condition, in some cases accounting for as many as 25–30% of patients with cryptococcosis [3,4]. Consequently, this clinical condition has become the focus of intense investigation because of the unique immunologic, epidemiologic, and therapeutic challenges these patients present to the clinician and investigator. The most devastating complication of cryptococcosis is cryptococcal meningoencephalitis (CME), which is associated with a high overall mortality that is highly variable depending on access to care and expertise in diagnosing and managing this condition. For example, 90-day mortality in people living with HIV (PLWH) as high as 70% is reported in resource-limited regions of the world with limited access to adequate care [5], whereas mortality in patients with and without immunocompromising conditions in regions with more widespread access to care and expertise in disease management may be as low as 10–20% [6,7].

Our cases are from a large regional referral center located in the Southeastern United States, which is highly endemic for cryptococcosis. Consequently, we encounter large numbers of patients with cryptococcosis, of which approximately 15% are phenotypically normal hosts, including those with CME [6]. Herein, we will review recent observations pertaining to normal hosts with CME and also share a more than 20-year experience at our institution, reviewing the outcomes of these normal hosts compared to those with CME associated with more traditional risk factors. This brief review will highlight some of the key clinical, epidemiologic, diagnostic, and therapeutic differences between phenotypically normal patients with CME and those with more traditional risk factors.

## 2. Microbiology

Fungi belonging to the genus Cryptococcus are basidiomycetes that are encapsulated yeasts. *Cryptococcus neoformans* and *Cryptococcus gattii* are the chief pathogens among humans, and inhalation is the usual route of primary infection. *Cryptococcus neoformans* was originally classified into serotypes A, B, C, D, and AD based on capsular agglutination reactions. More recently, it has been proposed that *C. neoformans* be divided into at least two subspecies (*C. neoformans* and *C. deneoformans*), while *C. gattii* has been divided into five subspecies (*C. gattii*, *C. bacillosporus*, *C. deuterogattii*, *C. tetragattii*, and *C. decagattii)* [8]. For practical purposes and for purposes of this review, we will refer simply to *Cryptococcus neoformans* and *Cryptococcus gattii*.

These two pathogens differ with respect to epidemiology, clinical manifestations, and geographic distribution [9,10]. *Cryptococcus neoformans* is a ubiquitous pathogen found in most temperate regions of the world [1,2]. It is most commonly found in decaying organic matter in many soil types, particularly that which has been enriched by animal and bird droppings. By contrast, *Cryptococcus gattii* has been found mainly in tropical and subtropical regions [9] but has also been identified in British Columbia and the Northwestern United States in conjunction with a well-documented outbreak first described in 1999 [11]. Even more recent data suggest that certain genotypes of *Cryptococcus gattii* are increasing outside of the Northwestern United States, with particular predominance in the Southeastern US [12]. Both major species of Cryptococcus have similar growth requirements on artificial media, but *C. gattii* can be readily differentiated from *C. neoformans* by plating the isolate on canavanine–glycine–bromothymol (CGB) agar. CGB agar turns blue in the presence of *C. gattii* but remains yellow with *C. neoformans*. Additionally, matrix-assisted laser desorption ionization-time-of-flight mass spectrometry, and the GenMark eplex blood culture identification panel are now more readily available to aid in rapidly differentiating between *C. neoformans* and *C. gatii*.

## 3. Epidemiology

Cryptococcosis is primarily an opportunistic infection, and it is usually seen among patients with cell-mediated and/or humoral immune disorders. Globally, there are approximately 223,000 cases of cryptococcosis annually, and there are an estimated 180,000 deaths due to HIV-associated CME [13]. The incidence of cryptococcosis has increased in the last 40 years, influenced by the HIV pandemic and the availability and broader use of newer immunosuppressive agents and biologic immunomodulators.

Primary infection with cryptococcus occurs primarily through inhalation of aerosolized propagules [1]. Current understanding suggests that most primary infections are asymptomatic. Based on pooled pediatric serologic data from three distinct regions, it can be assumed that a large proportion of adults worldwide have been previously exposed to this pathogen [14]. It follows that most clinical infections, particularly those resulting in meningoencephalitis, result from the reactivation of a latent cryptococcal infection or reinfection. Occasional primary infection occurs via either direct inoculation, such as trauma from pet birds, or presumed exposure from penetrating trauma, such as a wood splinter or thorn [15,16]. Human-to-human transmission does not occur except through accidental exposure to contaminated donor solid organs or corneal tissue following a transplant procedure [17,18,19]. Unfortunately, there is not a routinely available screening assay or skin test to determine previous exposure, which could greatly enhance the understanding of the natural history of cryptococcal infections.

While the majority of symptomatic infections occur among individuals with known underlying disorders of immune dysfunction, in many surveys, those with cryptococcosis who have no significant underlying disorder constitute up to 30% of the total among patients with symptomatic infections [3,4]. This is recognized more frequently in middle- and high-income countries, possibly as a function of more accurate reporting and a more diverse population composed of individuals with a variety of underlying disorders. For instance, in more developed parts of the world with greater access to more sophisticated medical care, rates of cryptococcosis have fallen among PLWH, but the disease has remained quite common among solid organ transplant recipients, those receiving chronic glucocorticosteroids, and those receiving biologic agents. Across all risk groups, those greater than 18 years old, male gender, and African American descent seem to be overrepresented among symptomatic patients [3,20]. Interestingly, diabetes mellitus has never been clearly defined as a risk factor for developing CME. However, limited data suggest that those with CME and diabetes alone have poorer outcomes than their otherwise normal counterparts [21].

Environmental exposure focusing on specific outdoor activities has led to little insight as it relates to the risk of developing cryptococcal disease. Specifically, only time spent outdoors in general seems to be a risk factor for disease acquisition, whereas more specific activities such as recent excavation, construction, hunting, and hiking are more difficult to definitively relate to developing symptomatic disease [22]. Consequently, there have been no specific recommendations for behavioral interventions aimed at the prevention of symptomatic cryptococcosis.

Finally, there are significant differences between *C. neoformans* and *C. gattii* as it relates to host factors. Specifically, *C. neoformans* tends to occur more commonly in those who have immune compromise, whereas *C. gattii* is diagnosed more often in otherwise normal individuals [9,10].

## 4. Clinical Features

The clinical manifestations of CME in the phenotypically normal host are not dissimilar from the traditional immunocompromised host. Headache, confusion, and altered mental status are the most common symptoms, whereas fever and meningismus are often absent, occurring in fewer than half of these patients [1,6,23]. This often presents a challenge to prompt diagnosis in these patients, especially among primary care providers where the diagnosis of CME in the normal host is considered infrequent. Indeed, our previous observations have suggested that the delay in diagnosis in non-HIV, nontransplant individuals with CME is a median of approximately 70 days from the onset of symptoms to diagnosis. For those with no known underlying host immune deficiency, the median time to diagnosis is almost 90 days [6,20]. At our center over the last two decades, we have encountered several individuals with CME whose onset of symptoms preceded diagnosis by 9–18 months.

Late complications of CME include focal neurologic findings such as altered vision, including diplopia and blindness, loss of hearing, and altered vestibular function as the most common manifestations [1,2,24]. It has been suggested that these complications may be more common in the normal host, in part due to a delay in diagnosis [23,25]. Indeed, it is often the recognition of these grave complications that leads the clinician ultimately to the diagnosis of CME in an individual who might have been suffering with progressive headaches, visual disturbances, hearing loss, imbalance, and/or confusion for weeks or months. Neurologic symptoms associated with CME can be related to direct invasion of brain parenchyma with the organism, resulting cerebral edema, and/or increased intracranial pressure with or without ventriculomegaly [23,26]. While these tend to be later complications in patients with more prolonged symptoms, they can be devastating if not recognized and managed promptly.

Several observational studies have now demonstrated that *C. neoformans* and *C. gattii* exhibit a different propensity for organ involvement and long-term complications based on host factors [10,24]. For example, most observational data suggest that *C. neoformans* is more likely to occur in immunocompromised individuals and has a greater propensity to cause meningoencephalitis, whereas *C. gattii* has a greater propensity to cause disease among normal hosts and is much more likely to cause isolated pulmonary disease. In the largest observational study to date, including 457 patients with *C. neoformans* and 257 with *C. gattii* infections, 10% and 40%, respectively, were seen in normal hosts [10]. CME in patients with and without underlying disease was identified in 65% and 50% of patients with *C. neoformans* and *C. gattii*, respectively [10]. Given that infections due to *C. neoformans* are more common in general, it is clearly a much more common cause of CME in this group.

## 5. Diagnosis

Prompt consideration of a diagnosis of CME is key to optimizing a favorable outcome. As noted above, prolonged delay in diagnosis is typical due to the lack of suspicion on the part of the clinician for this opportunistic pathogen occurring in non-HIV-infected hosts, and this can result in unfortunate consequences [27]. As such, it is critical to consider cryptococcosis as a cause of persistent neurologic symptoms in the normal host such as headache, visual difficulties, hearing loss, and imbalance, even in the absence of fever and meningismus, when other more common explanations have been reasonably excluded.

Once it is considered, the diagnosis of CME is straightforward due to the ready availability of the highly specific serum and CSF cryptococcal antigen (CrAg) assay. The current assay of choice is the CrAg lateral flow assay (LFA) due to its sensitivity, specificity, ease of use, and low cost [28]. This assay is positive in the CSF of approximately 95% of individuals with CME and is more sensitive than either routine culture or India ink among both HIV and non-HIV-infected individuals [28,29,30]. By comparison, cerebrospinal fluid (CSF) culture is positive in about 80% and India ink in about 50% of these patients. Associated findings in the CSF typically include lymphocyte predominant CSF pleocytosis, hypoglycorrhachia, and an elevated CSF protein [1]. The cryptococcal antigen LFA is also a useful screening assay in the serum when the performance of a lumbar puncture is contraindicated or not feasible, but it is not as sensitive when compared to the CSF assay.

Neuroimaging is an important part of the initial and follow-up evaluations of these patients. Focal parenchymal lesions are common with cryptococcal meningoencephalitis and are seen more commonly in normal hosts with *C. gattii* than *C. neoformans* [10]. Hydrocephalus and cerebral edema are critically important findings that may require intervention to prevent permanent neurologic sequelae and death. Other focal imaging findings, such as stroke, are less commonly seen in normal hosts. There are no studies that clearly demonstrate changes in neuroimaging findings among normal hosts and those with underlying immunosuppression.

## 6. Host Immunologic Evaluation

An evaluation of host immune function among patients who are phenotypically normal and present with CME should be considered. At a minimum, T-lymphocyte assays including CD3, CD4, and CD8 should be performed in an effort to rule out idiopathic CD4 lymphopenia (ICL) [31]. This is usually defined as chronic CD4 lymphopenia < 300 cells/mm^3^ or less than 20% of total lymphocytes, with no evidence of HIV or other explanation for these findings. In addition, obtaining quantitative immunoglobulins to rule out common variable immunodeficiency (CVID), hyper-IgE syndrome, and other antibody disorders is reasonable [32,33]. More recently, investigators have linked the risk of CME to the presence of autoantibodies to GM-CSF and IFN-γ in the otherwise normal host [34,35]. Occasional cases have been associated with immune disorders such as GATA2 deficiency, Job’s syndrome, chronic granulomatous disease (CGD), X-linked CD40 ligand deficiencies, and sporadic monocytopenia [36,37,38]. Obtaining T-lymphocyte assays and immunoglobulins are tests that should be performed at the time of the diagnosis of CME and can be completed in an inpatient or outpatient setting. We suspect that most patients who are considered phenotypically normal have an undiagnosed underlying immune dysfunction. Patients who initially present with cryptococcosis without any underlying immune disorders are considered phenotypically normal for the purposes of this paper, even after the above workup has been performed.

## 7. Antifungal Treatment

There are no recent randomized treatment studies comparing therapeutic approaches among phenotypically normal patients with CME. The only studies to enroll substantial numbers of patients without significant underlying disease were completed over 35 years ago [39,40]; virtually all subsequent randomized trials have been conducted almost exclusively among PLWH with CME [41]. Moreover, no therapeutic trial of extra-neural cryptococcosis has ever been conducted, so all current treatment recommendations for normal hosts with CME are derived almost exclusively from the previously mentioned clinical trials combined with the collective experience of transplant recipients and other immunocompromised patients with CME. Based on current published guidelines, patients with CME should receive at least 2 weeks of combination therapy with a lipid formulation of amphotericin B (3–5 mg/kg/d) plus flucytosine (100 mg/kg/d as four divided doses) for a minimum of 2 weeks [42]. For induction therapy in resource-limited settings where flucytosine is not readily available, a combination of amphotericin B deoxycholate plus fluconazole 800–1200 mg daily for 2 weeks could be used as an alternative. If amphotericin B deoxycholate is not readily available, fluconazole 1200 mg daily plus flucytosine 100 mg/kg daily can also be used [23]. Some have advocated for a longer course of “induction” therapy with 4–6 weeks of combination therapy with these agents, particularly in transplant recipients and those with *C. gattii* infections, reflecting the standard approach in the 1970s and 1980s [23,40]. Recent modifications to the approach in phenotypically normal patients with CME are strictly informed by anecdotal experience in small series [23,43]. Following successful induction therapy with an amphotericin B-based regimen, most practitioners step down to “consolidation therapy” with fluconazole 800 mg daily (or the equivalent based on renal function) for 8 to 10 weeks. This is followed by maintenance fluconazole at 200 to 400 mg daily for up to a year [42]. Although not based on firm data, the length of therapy is generally determined by clinical and mycologic (culture) factors, independent of the underlying disease, but the total duration of therapy can generally be limited to 12 months if the underlying condition is stable. There are a few published alternatives to this standard, although many practitioners will substitute a different azole for fluconazole, such as voriconazole, posaconazole, or even isavuconazole, if, in the opinion of the clinician, there is clinical justification for switching to one of these agents [23,44,45]. In the majority of normal patients with CME, antifungal therapy can be safely discontinued after 6–12 months, provided that there is stability or complete resolution of symptoms attributable to cryptococcal infection, CSF culture negativity, and a significant decrease in CSF CrAg, if this is available. For most individuals, a “test of cure” lumbar puncture is not performed at the end of therapy if the patient is doing well clinically [42]. Many immunocompromised patients with CME will require more prolonged therapy due to persistent symptoms or other concerns, but there are no specific guidelines to guide these decisions, and they are left to the discretion of the clinician.

## 8. Intracranial Pressure Management

To prevent pressure-related neuropathology such as cranial nerve abnormalities, cognitive disorders, and other permanent neurologic sequelae, it is important to maintain intracranial pressure within the normal range whenever possible. There are several acceptable approaches to intracranial pressure management. Increased intracranial pressure presents commonly in both normal and immunocompromised hosts, affecting about 50% of patients in each group. The first principle is that opening pressure must be measured with each lumbar puncture (LP). Persistently elevated opening pressure greater than 250 mm H_2_O should lead to the removal of enough CSF such that the closing pressure is at least half of the original opening pressure or within the normal range (less than 200 mm H_2_O) [42,46,47]. Lumbar puncture may be repeated daily or every other day until documented normal or persistently less than 250 mm H_2_O. There are only practical limits to the number of therapeutic LPs performed on an individual, usually determined by patient tolerance and the willingness to continue undergoing the procedure. A lumbar drain may be placed in lieu of repeat lumbar punctures. A permanent ventricular shunt is an option for those not responding to less invasive measures for intracranial pressure control. The timing of this is completely dependent on the clinician and resource availability. It is notable that permanent ventricular shunts are more likely among normal hosts than immunocompromised hosts with CME. In many parts of the world where permanent ventricular shunts are unavailable, repeat lumbar punctures are performed for as long as necessary to reduce increased intracranial pressure. There appears to be little role for dexamethasone, acetazolamide, or other pharmacologic attempts to reduce intracranial pressure [42].

## 9. Post-Infectious Inflammatory Response Syndrome

Unique to the phenotypically normal patient with CME is the potential development of a unique syndrome referred to as post-infectious inflammatory response syndrome (PIIRS), which occurs after the initiation of effective antifungal therapy in these patients [23,26,48,49]. Since immune reconstitution inflammatory syndrome (IRIS) in its classic form does not occur in these patients, this syndrome does represent an immunologic response to successful antifungal therapy, which can lead to harmful post-infectious inflammation in the intracranial space [26]. Table 1 describes clinical and laboratory criteria for PIIRS [26]. While it is beyond the scope of this paper to discuss the mechanistic features of PIIRS in detail, it is important to recognize that the development of PIIRS in patients can lead to persistent and evolving neurologic abnormalities such as hydrocephalus, focal cranial nerve abnormalities such as blindness, hearing loss and/or vestibular function, cognitive decline, and other persistent symptoms such as headache [50,51,52,53]. Unfortunately, there are no current diagnostic tests to confirm PIIRS, but the clinician must have a high index of suspicion to potentially prevent these detrimental complications.

Intervention with glucocorticosteroids has been suggested to be beneficial for patients with PIIRS, but there is only one small prospective study examining the benefits of therapeutic glucocorticosteroids to manage symptoms in this condition [54,55]. In our own limited experience, we have successfully managed five patients with adjunctive glucocorticosteroids to treat PIIRS clinically manifested as arachnoiditis, cauda equine syndrome, and worsening encephalopathy. In the absence of a prospective randomized trial, it has been hypothesized that glucocorticosteroids initiated early in the antifungal treatment process can abrogate this immunologic response, thus limiting harmful inflammation while not influencing the ability to clear the organism and cure the infection [49,55]. A prospective trial randomizing individuals to intervention with pre-emptive glucocorticosteroid therapy versus placebo is currently under consideration. No specific guidelines exist on the exact duration of treatment of PIIRS with glucocorticoids, though current studies support continuing glucocorticosteroids for weeks to months, depending on the clinical response of the patient. There is no known data to date regarding the treatment of PIIRS refractory to glucocorticosteroids. Further research is needed to explore alternative therapies for PIIRS.

## 10. Outcomes

Compared to a composite of all immunocompromised patients with CME, phenotypically normal patients have overall better survival at 30, 90, and 360 days [3,6,20]. This is best demonstrated in Figure 1, which depicts almost 25 years of observational data from our institution (1996–2020) (non-published). Among 270 immunocompromised patients with CME, 70 (24%) died within one year of follow-up, whereas among 40 phenotypically normal hosts with CME, only 7 (17%) died at this timepoint. Beyond mortality, quality of life is a critically important measurable outcome in these patients. It is quite likely that normal individuals experience more permanent long-term neurologic sequelae, such as cranial nerve abnormalities, when compared to their immunocompromised counterparts [23,55]. This is likely due to a relatively robust inflammatory response, higher opening pressures, and a prolonged delay in establishing a diagnosis, leading to more permanent cranial nerve and cognitive abnormalities [52].

## 11. Conclusions

Phenotypically normal individuals with CME present a number of significant clinical and therapeutic challenges. First, it has proven difficult to diagnose these individuals in a timely manner, with an average time to diagnosis of almost 3 months for normal hosts compared to 20 to 30 days for PLWH and transplant recipients. This delay in diagnosis clearly has an impact on the ultimate neurologic outcome in these patients. Second, treatment for this group of patients is undefined by recent randomized and retrospective data. Current treatment algorithms for CME are all based on approaches to HIV patients in particular and immunocompromised patients in general. Unfortunately, there are too few of these patients with CME to efficiently conduct a randomized control trial comparing different approaches to treatment. Intracranial pressure-related phenomena, including the need for permanent ventricular shunts, seem to be greater in this group than compromised hosts and again may relate to delayed diagnosis and/or PIIRS. Finally, once the patient with CME is receiving effective antifungal therapy, treatment with adjunctive glucocorticosteroids should be considered to mitigate the effects of PIIRS. This intervention has been demonstrated to be successful in small observational studies and should be considered among patients who demonstrate evidence of a post-infectious inflammatory syndrome following the initiation of effective antifungal therapy. A specific treatment regimen is not clearly defined, but an effective course would likely be given over several weeks or months. Overall, the management of these patients remains very challenging and presents a fertile area for future investigation, particularly in the realm of early diagnosis and early intervention with antifungal therapy, potentially combined with targeted immunotherapy for selected individuals in order to prevent post-infectious inflammatory complications.

## Figures and Tables

**Figure 1 pathogens-12-01303-f001:**
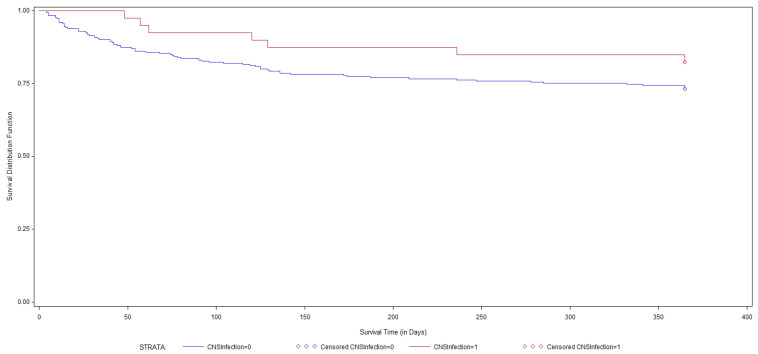
KM survival curve for normal (red line) vs. immunocompromised (blue line) patients with CME.

**Table 1 pathogens-12-01303-t001:** Clinical and laboratory definition of post-infectious inflammatory response syndrome (PIIRS) [26].

Main Criteria
1. Unchanged or declining mental status
2. Non-refractive visual defects
3. Hearing changes in a previously healthy patient with CSF fungal culture conversion after initial therapy with amphotericin B.
Supportive criteria
1. Elevated CSF WBC and protein, reduced CSF glucose
2. Increased CSF inflammatory markers such as IL-6 and soluble CD25 levels
3. Elevation in CSF activated immune cells (CD4, CD8 lymphocytes, natural killer cells, monocytes)
4. Abnormal brain and spinal cord MRI findings on post-contrast FLAIR showing leptomeningeal enhancement, choroid plexitis, ependymitis, evolving parenchymal lesions, hydrocephalus, and arachnoiditis

## Data Availability

The data presented in this study are available on request from the corresponding author.

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
