# Peer review of "Cryptococcal Meningoencephalitis in Phenotypically Normal Patients"

_pathogens, 2023, doi:10.3390/pathogens12111303_

Round 1

Reviewer 1 Report

Comments and Suggestions for Authors

The authors review literature describing the differences in cryptococcal meningoencephalitis in immunodeficient and ‘immunocompetent’ hosts.  In the current work, the authors use the term ‘phenotypically normal’ to describe patients with ‘no clinically obvious predisposition to disease.’  However in a previous review, ‘phenotypically normal’ patients were described as a ‘heterogeneous population that ranges from apparently normal hosts to those with significant immunologic impairment including those on chemotherapy and/or immunosuppressive therapy, those with organ dysfunction and those with innate or acquired immunodeficiencies’ (https://www.ncbi.nlm.nih.gov/pmc/articles/PMC3715903/).  As such, it is unclear if the authors are focusing the current review on 1) patients who do not have a history of HIV or organ transplant, or 2) patients without immunosuppressive risk factors for CM. Clarification is needed to better describe ‘phenotypically normal’ in relation to human genetic factors that predispose patients to cryptococcosis and CM.  Additionally, much of the data reported throughout the manuscript is derived from patients with HIV who have CM.  While this is likely due to the lack of data in patients who do not have a history of HIV or organ transplant (which the authors acknowledge), the authors should be more transparent and attribute these data to the respective populations (patients with HIV vs patients who do not have a history of HIV or organ transplant). The authors must present additional, much needed, details before this manuscript could be considered worthy of publication.

Abstract

·       L16: CME not defined

Introduction

·       L35-37: consider rewording

·       L40: consider a different descriptor rather than ‘easier’

·       L38-42: are these data from patients with HIV or patients who do not have a history of HIV or organ transplant? Notably, Jarvis 2014 enrolled patients with HIV and CM.

·       L43: define ‘ours’. Is this referring to the authors’ institution?

·       L45-46: please include a reference to support this statement

Microbiology

·       L67-70: please acknowledge recent data suggesting longer endemicity of C. gattii, specifically VGI-SE clade, in the SE USA (https://pubmed.ncbi.nlm.nih.gov/24019979/, https://pubmed.ncbi.nlm.nih.gov/27191335/)

Epidemiology

·       L79: please use the more recent analysis describing the global burden of CM in patients with HIV (https://pubmed.ncbi.nlm.nih.gov/36049486/) rather than the older version which is currently used throughout the manuscript

·       L86: delete ‘data’

·       L107: please use inclusive and non-stigmatizing language and avoid ‘HIV-positive persons’ (https://academic.oup.com/cid/article/76/10/1860/7016316)

·       L110: again, please use inclusive language and avoid ‘black race’ (https://jamanetwork.com/journals/jama/fullarticle/2783090). Perhaps the authors can address racism, disparities, and inequities in relation to risk factors for CM and how these might have contributed to overrepresentation of minority groups. Additionally, the authors could consider discussing whether these sociodemographic factors and social determinants are different among ‘phenotypically normal’ patients with CM.

Clinical features

·       L137-139: please include a reference to support this statement and describe why diagnosis was so significantly delayed at the authors’ institution vs previously published data

·       L160-165: Baddley et al described 457 patients with C. neoformans and 257 patients with C. gattii, of which 10% and 41% had no underlying diseases (please see table 1, https://www.ncbi.nlm.nih.gov/pmc/articles/PMC8473583/). Can the authors clarify their statements? Were the 65% and 50% of patients with CME considered ‘phenotypically normal’?

Diagnosis

·       L176-178: the study validating CrAg LFA by Boulware et al was conducted in people with HIV (https://pubmed.ncbi.nlm.nih.gov/24378231/). Perhaps Hevey et al would be more appropriate given the purpose of this manuscript (https://pubmed.ncbi.nlm.nih.gov/32848037/). Overall, it is unclear if the authors are describing the performance of diagnostic techniques in patients with HIV or patients who do not have a history of HIV or organ transplant. The authors should be more transparent and attribute these data to the respective populations (patients with HIV vs patients who do not have a history of HIV or organ transplant).

Host immunologic evaluation

·       The authors do a great job describing the baseline immunologic workup that should be performed in patients with CM who do not have apparent risk factors.

·       Should this workup be performed in the acute setting or deferred to the outpatient setting?

·       If any of these immunodeficiencies are identified, are patients still considered ‘phenotypically normal’?

Antifungal treatment

·       L236: period needed after ‘clinically’

·       L235-239: the authors mention that most patients will not undergo a ‘test of cure’ lumbar puncture but then recommend CSF culture sterility and decreased CSF CrAg. Can the authors clarify their statements?

·       L239-242: can the authors provide guidance on durations of maintenance therapy individuals who do not have a history of HIV or organ transplant but have an underlying immunodeficiency?

PIIRS

·       L289-291: please include a reference to support this statement and describe the doses and durations of glucocorticoids. Notably, the authors do not propose a duration for glucocorticoids here, but mention ‘weeks to months’ in the conclusion (L335)

Figure 1 is rather small and should be resized. It appears that the KM curve describes the mortality difference between patients with CM and those without CM. Can the authors provide further details? Can the authors clarify their definition of ‘immunocompromised’? Does this include patients with HIV, those on immunosuppressive therapy, and/or those with underlying immunodeficiencies. Additionally, ‘normal’ should be defined. It is unclear why the authors elected to include unpublished data. Readers are unable to interpret and verify these data.

Conclusions

·       L320 and L323please use inclusive and non-stigmatizing language and avoid ‘HIV-positive persons’ (https://academic.oup.com/cid/article/76/10/1860/7016316)

References

·       L402-403: ‘+’ at the end of the citation

Reviewer 2 Report

Comments and Suggestions for Authors

The topic of this review on "Cryptococcal meningoencephalitis in phenotypically normal patients" is interesting and the findings of this manuscript enriched the data about CME associated the phenotypically normal host. This review will add our knowledge about the differences in the clinical presentation, management, outcome, and potential risk factors for the phenotypically normal host compared to immunocompromised persons. The manuscript is well written and can be accepted after minor revisions.

1) Line 14: “Cryptococcal meningoencephalitis” should be “Cryptococcal meningoencephalitis (CME)

2) In the section of Diagnosis, the authors did not describe whether these diagnostic tests or neuroimaging have the differences in CME associated the phenotypically normal host and immunocompromised host.

3) In the phenotypically normal population, in fact, some patients may still have immune abnormalities. How can we identify these patients, such as the deficiencies in GM-CSF due to autoantibodies may play a role in infection with C. gattii? This section needs more explanation.

4) PIIRS is an important part of cryptococcal meningoencephalitis in phenotypically normal patients. Although the authors mention that the discussion on the mechanistic characteristics of PIIRS is beyond the scope of this paper, it is important for the authors to provide a detailed discussion on the clinical manifestations, diagnosis, treatment, and research progress of PIIRS. In addition, the authors should explain whether there are other alternative possible treatments when therapeutic glucocorticosteroids for refractory PIIRS is ineffective.

Reviewer 3 Report

Comments and Suggestions for Authors

Please see attached review.

Reviewer 4 Report

Comments and Suggestions for Authors

Diagnosis:

I suggest the authors to add the part: “Both major species of Cryptococcus have similar growth requirements on artificial media, but Cryptococcus gattii can be readily differentiated from C neoformans by plating the isolate on canavanine-glycine–bromothymol (CGB) agar. CGB agar turns blue in the presence of C gattii but remains yellow with C neoformans.” In the diagnosis instead of Microbiology because most laboratories differentiate these species using CGB agar.

Antifungal treatment:

CM treatment depends on the patient underlying condition, such as HIV or transplant related disease. There are alternative protocols, according to the suggested literature (https://www.ncbi.nlm.nih.gov/books/NBK525986/). I suggest the authors could at least cite that there are alternative protocols that include voriconazole, AMB lipid formulations and only fluconazole (because 5-FC is not available everywhere).

For non-HIV patients, there is a clinical trial that is not recruiting yet, but it will evaluate the efficacy and safety of ABCD in the treatment of cryptococcal meningitis in non-HIV patients at week 4, the end of induction therapy, week 10 and the end of consolidation therapy (NCT05471063).

The paper should be reviewed by the authors because there are some words written together, such as on line 278.

Suggested literature:

https://www.ncbi.nlm.nih.gov/books/NBK525986/

Round 2

Reviewer 4 Report

Comments and Suggestions for Authors

The authors made the requested changes.